# Anti-β2-GPI Antibodies Induce Endothelial Cell Expression of Tissue Factor by LRP6 Signal Transduction Pathway Involving Lipid Rafts

**DOI:** 10.3390/cells11081288

**Published:** 2022-04-11

**Authors:** Gloria Riitano, Antonella Capozzi, Serena Recalchi, Daniela Caissutti, Agostina Longo, Vincenzo Mattei, Fabrizio Conti, Roberta Misasi, Tina Garofalo, Maurizio Sorice, Valeria Manganelli

**Affiliations:** 1Department of Experimental Medicine, “Sapienza” University of Rome, 00161 Rome, Italy; gloria.riitano@uniroma1.it (G.R.); antonella.capozzi@uniroma1.it (A.C.); serena.recalchi@uniroma1.it (S.R.); daniela.caissutti@uniroma1.it (D.C.); agostina.longo@uniroma1.it (A.L.); vincenzo.mattei@uniroma1.it (V.M.); roberta.misasi@uniroma1.it (R.M.); tina.garofalo@uniroma1.it (T.G.); valeria.manganelli@uniroma1.it (V.M.); 2Biomedicine and Advanced Technologies Rieti Center, Sabina Universitas, 02100 Rieti, Italy; 3Rheumatology Unit, Department of Clinical Internal, Anesthesiological and Cardiovascular Sciences, “Sapienza” University of Rome, 00161 Rome, Italy; fabrizio.conti@uniroma1.it

**Keywords:** antiphospholipid syndrome, β2-GPI-glycoprotein I, LRP-6, Tissue Factor, signal transduction

## Abstract

In this study we analyzed whether anti-β2-GPI antibodies from patients with APS induce the endothelial cell expression of Tissue Factor (TF) by a LRP6 signal transduction pathway involving lipid rafts. HUVEC were stimulated with affinity purified anti-β2-GPI antibodies. Both LRP6 and β-catenin phosphorylation, as well as TF expression, were evaluated by western blot. Results demonstrated that triggering with affinity purified anti-β2-GPI antibodies induced LRP6 phosphorylation with consequent β-catenin activation, leading to TF expression on the cell surface. Interestingly, the lipid rafts affecting agent methyl-β-cyclodextrin as well as the LRP6 inhibitor Dickkopf 1 (DKK1) partially reduced the anti-β2-GPI antibodies effect, indicating that the anti-β2-GPI effects on TF expression may depend on a signalling transduction pathway involving both lipid rafts and LRP6. An interaction between β2-GPI, LRP6 and PAR-2 within these microdomains was demonstrated by gradient fractionation and coimmunoprecipitation experiments. Thus, anti-β2-GPI antibodies react with their target antigen likely associated to LRP6 and PAR-2 within plasma membrane lipid rafts of the endothelial cell. Anti-β2-GPI binding triggers β-catenin phosphorylation, leading to a procoagulant phenotype characterized by TF expression. These findings deal with a novel signal transduction pathway which provides new insight in the APS pathogenesis, improving the knowledge of valuable therapeutic target(s).

## 1. Introduction

“Antiphospholipid antibodies” (aPL) are a group of autoantibodies which, in association with thrombosis [arterial and/or venous] and/or pregnancy morbidity, characterize antiphospholipid antibody syndrome (APS) [1]. aPL are a heterogeneous family of antibodies reacting with phospholipid–protein complexes, among which β2-glycoprotein-I (β2-GPI) represents the main target antigen [2].

β2-GPI is a protein physiologically involved in the process of coagulation and its regulation. Thus, anti-β2-GPI antibodies may interfere with homeostasis and eventually contribute to the occurrence of thrombotic events [3]. Several studies demonstrated a key role for endothelial cells (ECs) in APS pathogenesis [4,5,6]. Indeed, it was suggested that ECs activation, occurring in a β2-GPI-dependent manner, may lead to cell dysfunction, resulting in increased risk of thrombosis, accelerated atherosclerosis, myocardial infarction and stroke in patients with APS [7,8,9]. Indeed, anti-β2-GPI antibodies are considered not only a serological marker of APS, but are directly involved in the pathogenesis of thrombosis, since they may interact with ECs and subsequently trigger an up-regulation of Tissue Factor (TF), the major initiator of the clotting cascade, inducing a procoagulant endothelial phenotype [10,11,12]. Several molecules have been reported as putative receptors of anti-β2-GPI antibodies, including Annexin A2, TLR-4, TLR-2, TLR-8, C3aR and Low-density lipoprotein receptor (LDLR)-related protein (LRP) 8 (LRP8) [13]. Raschi et al. [14] showed that the TLR-4 signaling pathway is activated by the anti-β2-GPI binding to ECs, with consequent phosphorylation of interleukin-1 receptor–associated kinase (IRAK) and the nuclear translocation of NF-kB. In previous studies we demonstrated that anti-β2-GPI antibodies react with their target antigen, likely in association with annexin A2 and TLR-4 within lipid rafts in the monocyte plasma membrane [15]. In particular, the anti-β2-GPI-triggered signal transduction pathway leads to a proinflammatory and procoagulant phenotype characterized by the release of TF [16,17].

Moreover, López-Pedrera demonstrated a correlation between protease-activated receptor-2 (PAR-2) levels and IgG aPL titers, as well as a parallel behavior of TF and PAR-2 expression [18]. Indeed, PAR-2 inhibition prevented the IgG aPL-induced TF expression, suggesting that the TF/PAR-2 axis is directly involved in the pathogenesis of the thrombotic complications associated with APS [7,18,19]. In addition, it was shown that TF signaling through PAR-2 mediates neutrophil activation and fetal death in APS [20]. Recently, LRP6, an important receptor belonging to the same family of LRP8, has been identified as a novel co-receptor of PAR-2, as revealed by coimmunoprecipitation experiments [21].

LRP6, a member of the LRP family, is a type I transmembrane receptor (C-terminus in cytosol) and an essential co-receptor of Wnt ligands for canonical β-catenin dependent signal transduction [22,23,24,25], thus playing a key role in numerous biological processes. LRP6 with a large extracellular domain binds several Wnt ligands [26], but also several Wnt antagonists, such as Dickkopf 1 (DKK1) and Sclerostin (SOST), which prevent Wnt–LRP6 binding and Frizzled–LRP6 complex formation [27,28,29].

In the absence of Wnt, β-catenin, which represents the key effector of this signaling pathway, is continuously degraded by the “degradation complex”; on the contrary, Wnt triggering through LRP6 dismantles the degradation complex, leading to the accumulation of unphosphorylated β-catenin. The stabilized β-catenin translocates to the nucleus and through transcription factors induces the expression of an array of genes downstream [21,30]. Haack et al. [31] suggested that lipid rafts are implicated in the Wnt/β-catenin pathway. A few papers indicated that lipid rafts may play a role in cell signaling involving LRP6 [32,33]. Recently, we showed the association of LRP6 with lipid rafts, demonstrating the key role of raft integrity in LRP6 cell signaling [26].

Since LRP6 would likely acquire a role in several protein-protein interactions, in this study we analyzed whether anti-β2-GPI antibodies from APS patients induce ECs expression of TF by a LRP6 signal transduction pathway involving lipid rafts.

## 2. Materials and Methods

### 2.1. Cell Culture and Treatments

Human umbilical vein endothelial cells (HUVECs) were maintained in PromoCell growth medium containing an endothelial cell growth medium kit (PromoCell, Heidelberg, Germany) and 10% fetal bovine serum (Sigma-Aldrich, Milan, Italy), at 37 °C in a humified 5% CO_2_ atmosphere. HUVECs (5 × 10^5^/mL) were seeded into 6-well cell culture and incubated at 37 °C for different incubation times with affinity-purified anti-β2-GPI antibodies (200 μg/mL), normal human IgG (200 μg/mL), or lipopolysaccharide (LPS) (100 ng/mL). In parallel experiments, HUVECs were pretreated with DKK1 (1 μg/mL) for 24 h, or with 5 mM methyl-β-cyclodextrin (MβCD, Sigma-Aldrich) for 30 min at 37 °C. After treatment, cells were collected and prepared for experimental procedures.

### 2.2. Purification of Anti-β2-GPI Antibodies

The isolation of human anti-β2-GPI IgG was obtained by affinity chromatography, as previously reported [14] from three APS patients (positive for anti-β2-GPI antibodies by ELISA). The three APS patients were women (ages 42, 42, and 44 years) with deep venous and arterial thromboses (Appendix A), who had been diagnosed according to the Sidney Classification Criteria [1]. The patients gave written informed consent, in compliance with the Helsinki Declaration. All of the three purified anti-β2-GPI IgG were positive at the dilution of 1:800, as detected by ELISA. As a control, we used IgG from normal human serum (Sigma-Aldrich).

### 2.3. Western Blotting Analysis

HUVECs, untreated or treated with affinity-purified anti-β2-GPI antibodies, normal human serum IgG or LPS, in the presence or in the absence of DKK1 or MβCD, were incubated for different incubation times (45 min and 4 h) at 37 °C, in 5% CO_2_. After treatments, the cells were resuspended in a lysis buffer, whose composition is 20 mM HEPES, pH 7.2, 10% glycerol, 1% Nonidet P-40, 50 mM NaF, 1 mM Na_3_VO_4_ and a protease inhibitors cocktail (Sigma-Aldrich). The lysates were centrifuged at 15,000× *g* at 4 °C for 15 min and soluble proteins were obtained. Protein quantitative analysis was determined by Bradford assay (Bio-Rad, Hercules, CA, USA). Samples were subjected to sodium dodecyl sulphate polyacrilamide gel electrophoresis (SDS-PAGE). Polyvinilidene difluoride (PVDF) membranes (Bio-Rad) were used to electrophoretically transfer the proteins and probed with rabbit anti-phospho-LRP6 Ab (R&D Systems, Minneapolis, MI, USA), rabbit anti-LRP6 mAb (Cell Signaling), anti-β-catenin mAb (Cell Signaling Technology), anti-phospho-β-catenin Ab (Cell Signaling Technology, Danvers, MA, USA), anti-TF mAb (Merck Millipore, Darmstadt, Germania). Horseradish peroxidase (HRP)-conjugated anti-rabbit IgG, anti-goat IgG or anti-mouse IgG (Sigma-Aldrich) were allowed to visualize the reaction. Immunoreactivity was assessed by using the ECL Western detection system (Amersham, Buckinghamshire, UK). A densitometric scanning analysis was performed by Mac OS X (Apple Computer International), using NIH Image J 1.62 software (National Institutes of Health; Bethesda, MD, USA)

### 2.4. Sucrose-Gradient Fractionation

Lipid raft fractions were isolated as previously described [34]. Briefly, 1 × 10^8^ HUVEC cells, untreated and treated as indicated, were lysed in 1 mL of homogenization buffer, containing 1% Triton X-100 (TX-100), 150 mM NaCl, 10 mM Tris-HCl (pH 7.5), 5 mM EDTA, 1 mM Na_3_VO_4_ and 75 U of aprotinin for 20 min at 4 °C. The lysate was first mechanically homogenized (10 strokes) and then centrifuged at 1300× *g* for 5 min. The supernatant fraction was placed at the base of a linear sucrose gradient (5–30%) and centrifugated in a SW41 rotor (Beckman Coulter, Inc, Palo Alto, CA, USA) at 200,000× *g* at 4 °C for 16–18 h. Eleven fractions were collected from the gradient, starting from the top of the tube. The fraction samples, loaded by volume, were analyzed by western blot. All steps were carried out at 0–4 °C.

### 2.5. Western Blot Analysis of Sucrose-Gradient Fractions

Fractions were subjected to 7.5% SDS-PAGE. The proteins were electrophoretically transferred onto PVDF membranes (Bio-Rad). Membranes were blocked with 1% BSA in TBS (Bio-Rad), containing 0.05% Tween 20 (Bio-Rad) and probed with rabbit anti-LRP6 mAb (Cell Signaling), with rabbit anti-phospho-LRP6 mAb (Cell Signaling), with rabbit anti-phospho-β-catenin mAb (Cell Signaling), with goat anti-β2-GPI (Affinity Biologicals, Ancaster, ON, Canada), with anti-Transferrin receptor, also known as Cluster of Differentiation 71 (CD71) mAb (Abcam, Cambridge, MA, USA), or with anti-flotillin polyclonal Ab (Abcam). The reaction was visualized with horseradish peroxidase (HRP)-conjugated anti-goat IgG (Sigma-Aldrich), anti-rabbit IgG (Sigma-Aldrich) or anti-mouse IgG (Sigma-Aldrich). The immunoreactivity was assessed by chemiluminescence by an ECL Western detection system (Amersham). A densitometric scanning analysis was performed with Mac OS X (Apple Computer International), using NIH Image J 1.62 software (National Institutes of Health; Bethesda, MD, USA).

### 2.6. Immunoprecipitation of LRP6

Pooled Triton X-100-insoluble fractions (4–5–6), from HUVECs cells, untreated or treated with affinity-purified anti-β2-GPI antibodies at 37 °C for 45 min, were immunoprecipitated with goat anti-LRP6 Abs (R&D Systems) [26]. Immunoprecipitation was also performed with an irrelevant polyclonal IgG, as a negative control. The immunoprecipitates were probed by western blot with rabbit mAb anti-LRP6 antibody (Cell Signaling), mouse anti-PAR-2 mAb (Millipore), or goat anti-β2-GPI (Affinity Biologicals Ancaster Canada).

### 2.7. Statistical Analysis

All of the statistical procedures were performed by GraphPad Prism software Inc. (San Diego, CA, USA). All data reported in this paper were verified in at least three different experiments performed in duplicate and reported as mean ± standard deviation (SD). *p* values for all graphs were generated using a student’s *t*-test as indicated in the figure legends; * *p* < 0.05, ** *p* < 0.005, *** *p* < 0.001, **** *p* < 0.0001.

## 3. Results

### 3.1. Anti-β2-GPI Antibodies Induce LRP6 and β-Catenin Phosphorylation

Western blot analysis showed that affinity purified human anti-β2-GPI antibodies, as well as LPS, induced LRP6 phosphorylation in HUVECs (Figure 1). On the contrary, when cells were stimulated with normal human IgG, virtually no anti-phospho-LRP6 reactivity was evident. Preincubating cells with MβCD, a lipid rafts affecting agent, or DKK1, a selective inhibitor of LRP6 signaling [27], almost completely prevented the anti-β2-GPI-triggered LRP6 phosphorylation, suggesting that lipid rafts integrity is required for anti-β2-GPI-mediated effect, leading to protein phosphorylation. Moreover, anti-β2-GPI antibodies, as well as LPS, induced β-catenin phosphorylation. Again, preincubating cells with MβCD or DKK1 almost completely prevented the anti-β2-GPI-triggered β-catenin phosphorylation, further supporting that both lipid rafts and LRP6 signaling are involved in the anti-β2-GPI-mediated effect. As a loading control, β-actin was employed. To exclude the possibility of LPS contamination, experiments were also carried out by preincubating cells with polymyxin B, which did not affect LRP6, or β-catenin phosphorylation triggered by human anti-β2-GPI antibodies (data not shown).

### 3.2. Anti-β2-GPI Antibodies Induce Tissue Factor Expression

In anti-β2-GPI-stimulated ECs, as well as after treatment with LPS, TF expression was significantly increased, as compared to both untreated and normal human IgG stimulated cells. Western blot analysis, after incubation with anti-β2-GPI antibodies, revealed a significant increase of TF expression (*p* < 0.001) as compared to unstimulated ECs (Figure 2). Interestingly, MβCD prevented (*p* < 0.0001) and DKK1 partially reduced (*p* < 0.001) the anti-β2-GPI antibodies effect, indicating that the anti-β2-GPI effects on TF expression may depend on a signalling transduction pathway involving both lipid rafts and LRP6. As a loading control, β-actin was employed. Experiments were also carried out by preincubating cells with polymyxin B, which did not affect TF expression, following human anti-β2GPI antibodies treatment (data not shown).

### 3.3. β2-GPI and LRP6 Preferential Association with Lipid Raft Fractions

The distribution of β2-GPI, LRP6, phospho-LRP6 and phospho-β-catenin was evaluated in raft fractions of ECs. For this purpose, 11 fractions, which include both TX-100 insoluble (4–5–6) and TX-100 soluble (10–11) fractions, were recovered by sucrose gradient and then analyzed by western lot. As expected, in untreated cells LRP6 was present in TX-100 insoluble fractions corresponding to lipid rafts [26], but also in TX-100 soluble fractions; by contrast, under triggering conditions, LRP6 was mostly enriched in TX-100 insoluble fractions, as revealed by densitometric analysis (Figure 3). Interestingly, phosphorylated LRP6, as well as phospho-β-catenin, were present exclusively in these TX-100 insoluble fractions, following anti-β2-GPI triggering. The analysis of β2-GPI distribution indicated that the anti-β2-GPI triggering induced β2-GPI enrichment in TX-100 insoluble fractions, suggesting that lipid rafts represent the plasma membrane sites from which β2-GPI promotes the signalling cascade upon binding to its ligand. All fractions were identified using two specific markers, flotillin, specific for lipid rafts, and CD71, which is excluded by lipid rafts. As expected, we observed that flotillin was consistently enriched in rafts (TX-100-insoluble fractions 4–5). By contrast, CD71 was localized exclusively in TX-100 soluble fractions.

### 3.4. Anti-β2-GPI Antibodies Trigger β2-GPI-LRP6 Interaction within Lipid Rafts of Endothelial Cells

In order to analyze the possible interaction between β2-GPI and LRP6 within lipid rafts, TX-100-insoluble fractions from either treated or untreated with human anti-β2-GPI antibodies were immunoprecipitated with anti-LRP6 Abs, followed by protein G-acrylic beads. The analysis of the immunoprecipitates by western blot demonstrated that in unstimulated cells, β2-GPI was slightly associated with LRP6. On the contrary, after treatment with human anti-β2-GPI antibodies, a significant portion of β2-GPI became associated with LRP6, as revealed by densitometric analysis (Figure 4). Since LRP6 has been identified as a novel co-receptor of PAR-2, we also analyzed whether PAR-2 may be associated with β2-GPI–LRP6 complex under the same stimuli conditions. Western blot analysis demonstrated that PAR-2 clearly co-immunoprecipitated with LRP6, mostly under anti-β2-GPI antibodies triggering (Figure 4). These results suggest that the complex β2-GPI–LRP6–PAR-2 associates within lipid rafts. No reactivity was shown using IgG with irrelevant specificity (Figure 4). The immunoprecipitation was verified by western blot.

## 4. Discussion

These findings demonstrate that human anti-β2-GPI antibodies induce a proinflammatory and procoagulant EC phenotype, characterized by the expression of TF, by triggering a signal transduction pathway involving LRP6 and β-catenin through lipid rafts (Figure 5).

In previous studies it was shown that anti-β2-GPI antibodies from APS patients induce IRAK serine phosphorylation and consequent NF-kB activation in endothelial [14] and monocyte cells [15]. We demonstrated a key role for lipid rafts in triggering this signaling pathway. In fact, anti-β2-GPI antibodies were unable to induce IRAK phosphorylation in the presence of MβCD, a well-known raft-disrupting agent [15].

In the present study we demonstrated for the first time the involvement of an additional signal transduction pathway in the procoagulant effect of anti-β2-GPI antibodies. Indeed, following the observations that PAR-2 inhibition prevented the aPL-induced TF expression [19] and that LRP6 has been identified as a novel co-receptor of PAR-2 [21], we demonstrated that anti-β2-GPI trigger LRP6 signaling, with consequent β-catenin phosphorylation. This additional signaling pathway appears to be also involved in the procoagulant phenotype of ECs, since we observed that anti-β2-GPI-induced TF expression was significantly, but not completely, reduced by treatment with DKK1, a selective inhibitor of LRP6. This finding is not surprising, since LRP6 is an important receptor belonging to the same family of LRP8, which has already been shown as a putative receptor of anti-β2-GPI [3]. Moreover, the anti-β2-GPI-induced TF expression was prevented by treatment with MβCD, indicating that the signal transduction pathway involving LRP6 and β-catenin is activated through lipid rafts. In this regard, a further indication derives from the observation that anti-β2-GPI target antigen, as well as phosphorylated LRP6 and phospho-β-catenin, were present within plasma membrane lipid rafts [35], as revealed by sucrose gradient analysis in the presence of Triton X-100. Lipid rafts are microdomains of cell membranes, characterized by greater rigidity compared to the rest of the membrane thanks to their composition enriched in cholesterol and sphingolipids. They show an essential role in the assembly of protein complexes involved in cell signaling [35,36,37] and in several pathophysiological mechanisms [38,39,40], including cell apoptosis [41,42].

Our findings suggested that anti-β2-GPI antibodies bind their receptor complex within lipid rafts. This hypothesis was strongly supported by the observation that LRP6 was highly enriched in lipid rafts, where it showed an association with β2-GPI, demonstrated by coimmunoprecipitation experiments which revealed β2-GPI coupled with LRP6. Moreover, since LRP6 has been identified as a novel co-receptor of PAR-2, we also analyzed whether PAR-2 may be associated with this complex, demonstrating that β2-GPI–LRP6–PAR-2 association occurs within lipid rafts. This immune complex might then induce cell signaling. The presence of PAR-2 in this complex is in agreement with the observation that a relationship between PAR-2 levels and IgG aPL titers was observed, and with the direct involvement of PAR-2 signaling in increased TF expression [18,43]. On the other hand, a direct PAR-2/LRP6 interaction has been recently demonstrated [21].

## 5. Conclusions

In conclusion, our study demonstrates for the first time that anti-β2-GPI antibodies bind their antigen closely in association with LRP6 and that this association occurs within the lipid rafts of the plasma membrane of ECs.

The identification of a new molecular complex involved in the binding of anti-β2-GPI antibodies with endothelial cells improves our knowledge of the receptor complex acting as a ligand of the antibodies on the cell plasma membrane. This observation fully agrees with the identification of oxidized phospholipids as ligands for LRP6, revealing that LRP6 serves as a specific receptor for oxidized phospholipids, which are products of lipid oxidation involved in pathological conditions such as atherosclerosis, inflammation, and APS [44].

Taken together, our results provide new insights into the pathogenesis of APS and introduce new hypotheses for valuable therapeutic targets, including LRP6, β-catenin and lipid rafts. In this regard, cyclodextrins could be used to reduce undesirable pharmaceutical characteristics or improve therapeutic indices and the site-targeted delivery of various drugs, including non-steroidal anti-inflammatory drugs [45]. Thus, some pharmacologic implications may derive from the results reported here, and the effects of lipid rafts affecting drugs, including statins, requires further study in APS patients.

## Figures and Tables

**Figure 1 cells-11-01288-f001:**
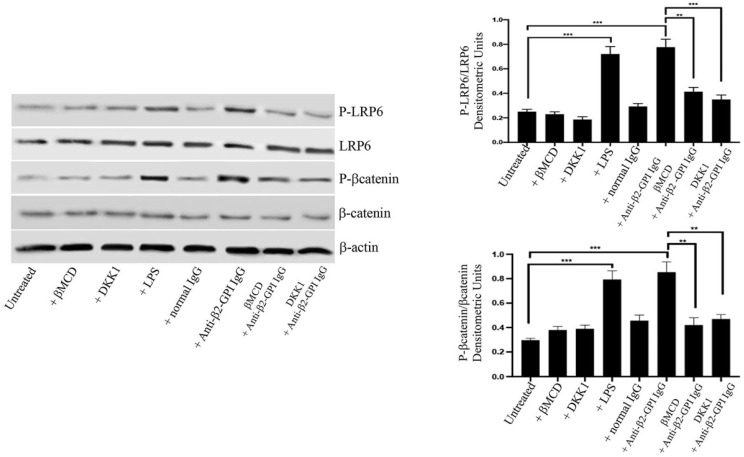
LRP6 and β-catenin phosphorylation following triggering with anti-β2-GPI antibodies. HUVECs, untreated or treated with affinity-purified anti-β2-GPI antibodies for 45 min, or with normal human serum IgG or LPS, in the presence or in the absence of DKK1 or MβCD, were analyzed by western blot using anti-phospho-LRP6 mAb, anti-LRP6 mAb, anti-phospho-β-catenin mAb or anti-β-catenin mAb. As a loading control, β-actin was employed. Densitometric analysis is shown. Results represent the mean ± SD from three independent experiments. ** *p* < 0.005, *** *p* < 0.001.

**Figure 2 cells-11-01288-f002:**
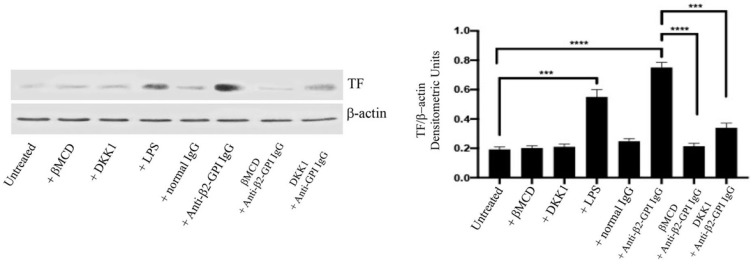
Tissue Factor expression following treatment with anti-β2-GPI antibodies. HUVECs, untreated or treated with affinity-purified anti-β2-GPI antibodies for 4 h or with normal human serum IgG or LPS, in the presence or in the absence of DKK1 or MβCD, were analyzed by western blot using anti-TF mAb. β-actin was employed as a loading control. Densitometric analysis is shown. Results represent the mean ± SD from three independent experiments. *** *p* < 0.001, **** *p* < 0.0001.

**Figure 3 cells-11-01288-f003:**
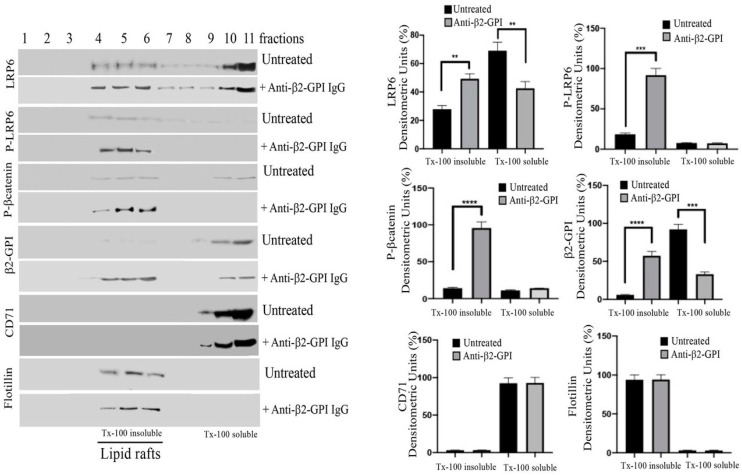
Lipid microdomains localization of LRP6 and β2-GPI following triggering with anti-β2-GPI antibodies. Sucrose gradient fractions obtained from HUVECs, untreated or treated with affinity-purified anti-β2-GPI antibodies for 45 min, were analyzed by western blot using anti-LRP6 mAb, anti-phospho-LRP6 mAb, anti-phospho-β-catenin mAb, anti-β2-GPI, anti-CD71 mAb or anti-flotillin Ab. Right panel. Bar graphs of densitometric analysis. The columns indicate the percentage distribution across the gel of raft fractions 4–5–6 (Triton X-100 insoluble fractions) and 9–10–11 (Triton X-100 soluble fractions), as detected by scanning densitometric analysis. Results represent the mean ± SD from three independent experiments. ** *p* < 0.005, *** *p* < 0.001, **** *p* < 0.0001.

**Figure 4 cells-11-01288-f004:**
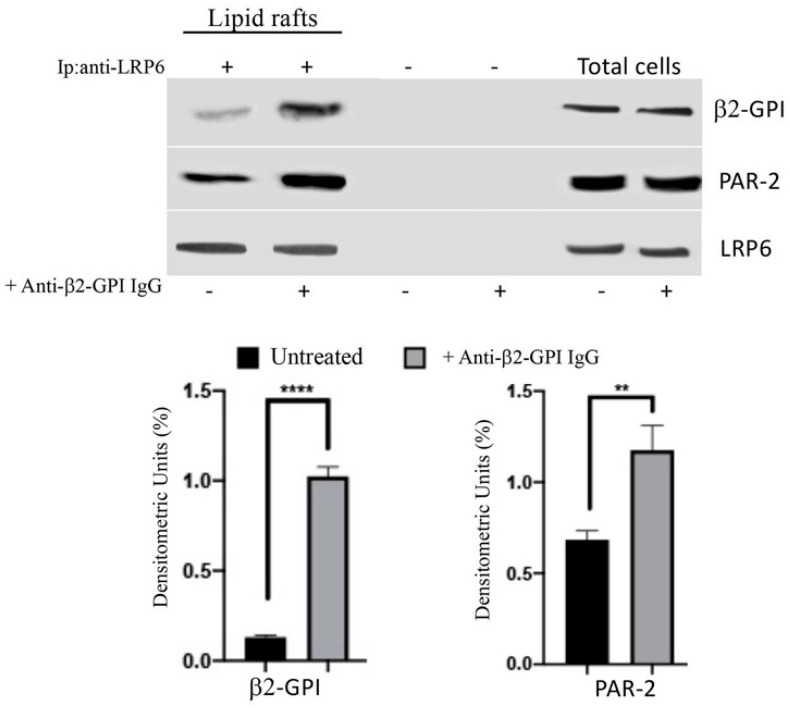
β2-GPI-LRP6-PAR-2 association within lipid rafts following triggering with anti-β2-GPI antibodies. Pooled Triton X-100-insoluble fractions (4–5–6), obtained from HUVECs cells, untreated or treated with affinity-purified anti-β2-GPI antibodies for 45 min at 37 °C, were immunoprecipitated with goat anti-LRP6 Abs. The immunoprecipitates were subjected to 10% SDS-PAGE. Membranes were probed with anti-PAR-2 mAb or with anti-β2-GPI Ab. As a control, immunoprecipitates were assessed by immunoblot with rabbit anti-LRP6 mAb. Results represent the mean ± SD from three independent experiments. ** *p* < 0.005, **** *p* < 0.0001.

**Figure 5 cells-11-01288-f005:**
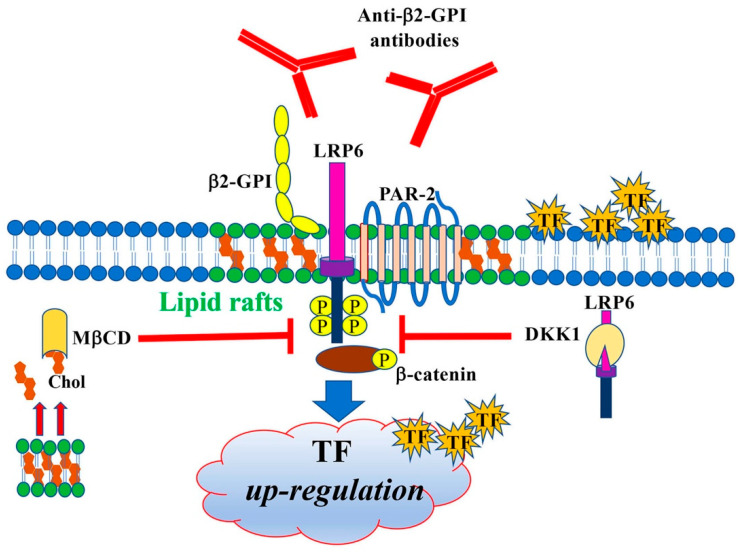
Signal transduction pathway triggered by anti-β2-GPI antibodies. Schematic drawing depicting the signal transduction pathway triggered by anti-β2-GPI antibodies, involving LRP6 and β-catenin phosphorylation, through lipid rafts. It leads to a procoagulant EC phenotype characterized by the expression of TF.

## Data Availability

The data underlying this article will be shared on reasonable request to the corresponding author.

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
