# Peer review of "Anti-β2-GPI Antibodies Induce Endothelial Cell Expression of Tissue Factor by LRP6 Signal Transduction Pathway Involving Lipid Rafts"

_cells, 2022, doi:10.3390/cells11081288_

Round 1
Reviewer 1 Report
It is an interesting research article examining whether anti-β2-GPI antibodies from antiphospholipid syndrome (APS) patients can induce the expression of Tissue Factor (TF) from human umbilical vein endothelial cells (HUVECs) by a LRP6 signal transduction pathway involving the lipid rafts. The authors demonstrated that triggering with purified human anti-β2-GPI antibodies could induce LRP6 phosphorylation with β-catenin activation, leading to TF expression on the surface of ECs. Furthermore, the lipid rafts affecting agent methyl-β-cyclodextrin and the LRP6 inhibitor could reduce the effects of anti-β2-GPI antibodies. Moreover, their study demonstrated, by coimmunoprecipitation experiments, that anti-b2-GPI antibodies react with the antigens strictly in association with LRP6 within the lipid rafts, indicating an additional signal transduction pathway in the procoagulant effect of such antibodies. Indeed, these findings provided new insight into the APS pathogenesis, introducing a new task for valuable therapeutic targets, including LRP6, β-catenin and lipid rafts.
The manuscript is well written in English and the contents are relevant to the potential clinical application. There are only minor suggestions as follows.
- In the Materials and Methods section 2.2., anti-β2-GPI IgG antibodies were purified from three APS cases, but there was only a brief description for these patients. The authors should add a Table to describe the clinical, laboratory and therapeutic profiles in these patients in detail.
- Since the positive results in Figures 1 to 4 were based on the isolated anti-β2-GPI antibodies, the authors should consider to measure the titers of their purified antibodies by standardization methods like enzyme-linked immunosorbent assay.
- There were some typographical errors in the manuscript. For example, in Result section 3.3., “β2-GPI” rather than “(2-.GPI”.
Author Response
The manuscript is well written in English and the contents are relevant to the potential clinical application. There are only minor suggestions as follows.
1. In the Materials and Methods section 2.2., anti-β2-GPI IgG antibodies were purified from three APS cases, but there was only a brief description for these patients. The authors should add a Table to describe the clinical, laboratory and therapeutic profiles in these patients in detail.
Authors: We added a new Table (S1), reporting the main clinical and laboratory profiles of the 3 patients, including the IgG anti-b2-GPI positivity (UA/mL).
2. Since the positive results in Figures 1 to 4 were based on the isolated anti-β2-GPI antibodies, the authors should consider to measure the titers of their purified antibodies by standardization methods like enzyme-linked immunosorbent assay.
Authors: We reported in the Materials and Methods section the titers of the purified anti-β2-GPI antibodies (1:800).
3. There were some typographical errors in the manuscript. For example, in Result section 3.3., “β2-GPI” rather than “(2-.GPI”.
Authors: We revised carefully the text.
Reviewer 2 Report
Very useful basic research. The results are excellently presented and justified.
The only comment is on a possible assessment of the thrombogenic risk of this interaction. Are you not considering expressing thrombin by a generation test that induces TF?
Author Response
Very useful basic research. The results are excellently presented and justified.
The only comment is on a possible assessment of the thrombogenic risk of this interaction. Are you not considering expressing thrombin by a generation test that induces TF?
Authors: We thank the reviewer for his/her positive comment.
In this study we focused on the signaling transduction pathway, which induce TF expression and release. We hypothesize that anti-b2-GPI antibodies may induce thrombin generation, since we previously demonstrated a procoagulant activity of TF, as revealed by Factor X-activating assay (Capozzi et al., J Thromb Haemost 19:2302,2021). It will be further analyzed in future studies.